# Effects of Biofilms on Trace Metal Adsorption on Plastics in Freshwater Systems

**DOI:** 10.3390/ijerph192113752

**Published:** 2022-10-22

**Authors:** Zhilin Liu, Tanveer M. Adyel, Zhiyuan Wang, Jun Wu, Jianchao Liu, Lingzhan Miao, Jun Hou

**Affiliations:** 1Key Laboratory of Integrated Regulation and Resources Development on Shallow Lakes, Ministry of Education, College of Environment, Hohai University, Nanjing 210098, China; 2Centre for Integrative Ecology, School of Life and Environmental Sciences, Deakin University, Melbourne, VIC 3125, Australia; 3Nanjing Hydraulic Research Institute, Nanjing 210098, China

**Keywords:** biofilms, trace metal, plastic debris, adsorption mechanism

## Abstract

The formation of plastisphere on plastics and their potential impact on freshwater ecosystems have drawn increasing attention. However, there is still limited information about the effects of plastisphere on the heavy metal adsorption capacity and the related mechanism of plastic debris in different freshwaters. Herein, the trace metal adsorption capacity, kinetics and adsorption mechanisms of virgin and biofilm-covered plastic debris were investigated. Polypropylene (PP) and polyethylene terephthalate (PET) plastic debris were placed in three freshwaters (Xuanwu Lake, Donghu Lake and the Qinhuai River) for 45 days to incubate biofilms. Batch adsorption experiments were performed to compare the adsorption processes of trace metal on virgin and biofilm-covered plastics. Results showed that biofilms increase the adsorption of metals on plastics, and the adsorption isotherms were well fitted by the Langmuir model. Furthermore, the adsorption capacities for lead (Pb(II)) were higher than that of cadmium (Cd(II)) and zinc (Zn(II)), with 256.21 and 277.38 μg/g (Pb(II)) adsorbed in biofilm-covered PP and PET, respectively, in Xuanwu Lake. The adsorption kinetics of metals on plastic debris were significantly affected by the biofilms, by switching the intraparticle diffusion for virgin plastic debris to film diffusion for the biofilm-covered plastic debris. Moreover, the complexation of functional groups within the biofilms might mainly contribute to the increases of metal adsorption, involving the participation of oxygen and nitrogen groups. Overall, these results suggested that biofilms reinforce the potential role of plastics as a carrier of trace metals in freshwaters.

## 1. Introduction

The global increase in the production, application and mismanagement of plastic has resulted in the widespread disposal of plastic debris in aquatic environments [1,2]. The increasing prevalence of these man-made plastics has drawn widespread public and regulatory attention [3]. The environmental behaviours of plastic debris and its biological effects on aquatic ecosystems have been extensively studied over the past decade [4,5]. Substantial evidence indicates that plastic debris can cause negative effects on aquatic organisms, threating the aquatic biodiversity and ecosystem health [6,7,8]. Meanwhile, because of its durability and mobility, plastic debris represents an important carrier for pollutants over long periods and distances, significantly affecting their transport fate in aquatic environments [9,10]. 

It has been widely documented in previous studies that trace metals could be highly adsorbed to plastic debris and the adsorption capacity of plastic debris varied with the properties of plastics, adsorbents and environmental factors [11]. For example, the adsorption of trace metals, i.e., copper, cadmium and lead, on plastic debris was reported by electrostatic interaction and surface complexation [12]. In the presence of iron, the adsorption amounts of metals (manganese (Mn (II)), zinc (Zn (II), Cu(II) and Pb(II) to plastic debris was reduced by 7.9% [13,14]. Besides, various environmental factors have been shown to alter the metal adsorption of plastic debris [14,15,16]. Furthermore, plastic debris aged by physicochemical impacts or the biofilms were reported to improve the trace metal adsorption [16,17,18]. Composed of a variety of microorganisms, biofilms can easily colonise the surface of plastic debris, forming the plastisphere [19,20]. Due to the ubiquity and complexity of biofilms, they played essential roles in the adsorption capacity of plastics, significantly affecting the environmental processes of trace metals in aquatic environments [11].

Studies have reported that the adsorption capacity of biofilm-covered plastic debris to trace metals was enhanced, due to the growth of biofilms altering the surface morphology and the physicochemical properties (surface charge and electrostatic interaction) of plastic debris [21]. The formation and development of biofilms on plastics reduced the surface hydrophobicity and promoted the generation of hydrous oxides, consequently enhancing the adsorption of trace metals on plastic debris [22]. Moreover, biofilms can be served as complex adsorption systems that could capture and accumulate various trace metals in aquatic environments. A previous study reported that compared to virgin plastics, the formation of biofilms on the surface of plastics increased the adsorption of Cu(II) and tetracycline though the film diffusion mechanism [23]. Thus, there is still a need to investigate the effect of biofilms on the mechanism of the trace metal adsorption capacity of plastic debris in different freshwaters [11].

In this study, polypropylene (PP) and polyethene terephthalate (PET) were selected to represent plastic debris. PP and PET are widely used and frequently detected in freshwaters. In this study, the objectives are (1) to compare the trace metals (Zn(II), Cd(II) and Pb(II)) adsorption capacity and kinetics of virgin PP, virgin PET, biofilm-covered PP and biofilm-covered PET; (2) to analyse the effects of the biofilm bacterial community on plastic debris adsorption behaviour; (3) to study the adsorption mechanisms by analysis of the changes of plastics’ surface properties in the aquatic environment.

## 2. Materials and Methods 

### 2.1. Plastics and Exposure Device

The plastic debris were obtained from household plastics—PET and PP from mineral water bottles and disposable lunch boxes, respectively purchased from Walmart Co., Ltd., Nanjing, China. The plastics were sliced into 10 cm × 10 cm pieces. According to our previous studies, 10 pieces of each type of plastics were placed vertically in a stainless-steel device (diameter 30 cm, mesh size 1 cm) by using fish line, which ensured that the plastics remained in full contact with the water [24]. 

### 2.2. Biofilm Incubation and Collection

We deployed 18 culture devices in total, with three replicates for each type of plastic at each exposure location. The devices were sited in Xuanwu Lake, Donghu Lake, and the Qinhuai River in Nanjing, China (Appendix A). As urban freshwaters, these sites experience the potential risk for plastic pollution releasing from a nearby area covering an industrial setting, residential area and traffic ways. The exposure devices were placed 1 m below the water surface, such that devices could experience sun light intensity and water fluctuations similar to our published studies [24,25]. Plastics-attached biofilm samples were collected and analysed after 45 days of exposure. Each plastic was rinsed three times with deionised water to remove unattached organisms. 

Approximately 0.5 g of biofilm from each sample were collected with sterile brush and sterile blade and placed into a sterile centrifuge tube. After collection, samples were stored at −80 °C for DNA extraction and plastics were retained in cooler and transported to the laboratory for further processing. 

The biofilm-covered plastics were sliced into 5 mm × 5 mm pieces and placed 10 pieces into a 100 mL centrifuge tube. After collection, samples were immediately stored at 4 °C to be used in the following experiments. For each type of plastic and each exposure location, samples were taken from three parallel devices.

After 45 days of exposure, the surface properties of biofilm-covered plastics were measured. The contact angle of plastic debris was measured by the optical video contact tester (JY-82B Kruss DSA, Hamburg, Germany). The surface morphology of plastics was observed by SEM (Hitachi S-4800, Tokyo, Japan) and the elements distribution on the plastic debris were determined by Energy dispersive spectroscopy (EDS). To analyse the change of chemical compositions, ATR-FTIR (Thermo Fisher Nicolet iS20, Allentown, PA, USA) was adopted [11]. The details were given in the Appendix A. 

### 2.3. DNA Extraction and 16S rRNA Sequencing

DNA was extracted using the E.Z.N.A.^®^ Tissue DNA kit (Omega Biotek, Norcross, GA, USA). The primers 515F (5′-GTGCCAGCMGCCGCGG-3′) and 806R (5′-GGACTACHVGGGTWTCTAAT-3′) were used to amplify the V4 region of the 16S rRNA gene for the bacterial communities [19,26]. The PCR products were purified using 1% agarose gel electrophoresis. Amplified libraries were established by NEBNext^®^ Ultra™ II DNA Library Prep Kit for Illumina^®^ (New England Biolabs, Ipswich, MA, USA). The libraries were sequenced on the Illumina Nova 6000 (Guangdong Magigene Biotechnology Co., Ltd., Guangzhou, China). 

The amplicons with sequences shorter than 200 bp and of low quality (quality score < 25) were removed from the raw sequence data. The clustering algorithm of OTUs was based on UPARSE. All sequences were subsampled to 19,160 sequences (the least value observed) and then used for further analysis. 

### 2.4. Batch Adsorption Experiments

In this study, adsorption isotherm experiments of Zn(II), Cd(II) and Pb(II) onto PP, PET, biofilm-covered PP and biofilm-covered PET were conducted with minimal mineral salts (MMS) solution, with a wide range of initial concentrations from 10 to 1000 μg/L. The composition of MMS solution is provided in Appendix A. A total of 10 pieces of PP, PET, biofilm-covered PP and biofilm-covered PET (all plastic sliced into 5 mm × 5 mm pieces) were introduced to 50 mL of the solutions and then shaken at 25 ± 1 °C for 48 h. The pH of the solution was maintained at 6.0 ± 0.1 using 0.01 mol/L HNO_3_ and NaOH. Then, the aqueous samples were collected after filtration through a 0.22 µm membrane filter and the concentrations were determined. The same experimental setups were performed for the kinetic experiments. Briefly, the original concentration of the three metals was set at 1000 μg/L, and then aqueous samples were collected with time intervals (0.5, 2, 4, 8, 12, 24 and 48 h). 

For the laboratory quality assurance and quality control, we added the control group (50 mL MMS solution without plastic debris and trace metals, and 50 mL MMS solution without plastic debris) in all the isotherm and adsorption kinetic experiments. The concentrations of three metals were measured with ICP-MASS (Agilent 7500, Santa Clara, CA, USA) in the Analytical Laboratory of the Jiangsu Provincial Center for Disease Control and Prevention. The details are given in the Appendix A.

### 2.5. Analysis of Adsorption Experiments

In this study, the adsorption isotherm and kinetic experiments were conducted in triplicate. The adsorption parameters and fitted curves for sorption isotherm and kinetic models were analysed by the nonlinear least-squares fitting or linear least-squares method, (OriginLab, Northampton, MA, USA). The details are given in the Appendix A. 

## 3. Results and Discussion

### 3.1. Characterisation of Plastic Debris

#### 3.1.1. Morphology of Plastic Debris

The surfaces of plastic debris exposed to aquatic systems were characterised by SEM (Figure 1). The images showed that compared with the virgin plastics, a large amount of biomass was observed from the surface of biofilm-covered plastic debris after 45 days of exposure, mainly consisting of algae and rod-like, palisade bacteria. The elements’ distribution on the plastic debris surfaces determined by the energy dispersive spectroscopy are shown in Appendix A. The two dominant elements on the surface of virgin plastic debris were C and O, which amount to >90% together. However, in the biofilm-covered plastic debris, the O element increased by 10 to 20% on the plastic debris surface, which may strongly affect its ability to adsorb metal ions [27,28]. Recent studies have demonstrated that the metal adsorption on plastic debris was enhanced by the biofilm covering, due to the oxygen-containing groups (C=O, C-O-C, C-O and O-C=O) binding of trace metals [29,30]. In addition, the formation and development of biofilms on plastic debris could reduce the surface hydrophobicity (Appendix A), which is beneficial for the adsorption of trace metals on plastics [22].

#### 3.1.2. Composition of Microbial Communities in Biofilms on Plastic Debris

Figure 2 shows distinct bacterial growth on plastic debris. The biofilm growth on the plastic debris followed the same trend in the three freshwaters. Proteobacteria, Cyanobacteria and Bacteroidetes were the three dominant phyla in the biofilm samples. Specifically, the microbial assemblages were dominated by Proteobacteria (44.28%) and Cyanobacteria (19.84%) for biofilm-covered PP, by Proteobacteria (41.37%) and Cyanobacteria (19.38%) for biofilm-covered PET in Xuanwu Lake. Proteobacteria (62.02%) and Bacteroidetes (9.75%) were the two dominant phyla for biofilm-covered PP and Proteobacteria (59.16%) and Cyanobacteria (9.26%) were dominant for biofilm-covered PET in the Donghu Lake. In the Qinhuai River, Proteobacteria (62.08%) and Bacteroidetes (12.54%) were dominant for biofilm-covered PP, and by Proteobacteria (63.42%) and Bacteroidetes (10.71%) for biofilm-covered PET.

Accordingly, Proteobacteria, Bacteroidetes and Cyanobacteria generally are the dominant phyla within natural biofilms [31]. Proteobacteria are always detected as early colonisers of plastic debris in freshwater [19]. Cyanobacteria are omnipresent in freshwater and can produce cyanobacterial toxins [32,33]. The formation of biofilms could significantly influence the morphology of plastic debris and diverse microbial communities on plastic debris may provide numerous binding sites for the adsorption of metals in aquatic environments [21]. 

#### 3.1.3. The Functional Groups of Biofilm-Covered Plastics

Figure 3 shows the FT-IR spectra of plastic debris with and without biofilms. Compared to virgin plastic debris, more peaks were observed in biofilm-covered plastic debris, corresponding to the functional groups within biofilms and the produced extracellular polymeric substances (EPSs) [34]. EPSs contain plenty of ionisable functional groups, including carboxyl, phosphoryl, amino and hydroxyl groups, which could increase the binding of metals [35].

The infrared spectra of the PP plastic debris showed that the absorption peaks of methyl appear at 2918 cm^−1^ and 1368 cm^−1^; the twisting deformation vibration of the methyl group overlaps with that of methylene and this absorption peak slightly shifts to 1458 cm^−1^, =C-H at 979 cm^−1^. Furthermore, for the biofilm-covered PP, -N=C=O at 2243 cm^−1^, C=O at 1888 cm^−1^ and 1660 cm^−1^ were also observed, respectively.

The infrared spectra of the PET plastic debris showed the position of the carbonyl-stretching vibration absorption peak shifts to 1717 cm^−1^ by the conjugation of the benzene ring, and the asymmetric stretching vibration of C-C-O containing the benzene carbon ring appears at 1243 cm^−1^. The peaks at 1094 cm^−1^ are absorption peaks of asymmetric stretching vibration of O-CH_2_-CH_2_-. The absorption peak of C-H rocking vibration on the benzene ring shifts to 719 cm^−1^, influenced by the carbonyl group. In addition, for the biofilm-covered PET, the peak at 2263 and 2181 cm^−1^ indicates that -N=C=O and -O=C=O, respectively.

Accordingly, both the C=O and -N=C=O peaks can be served as electron-donating groups to take part in the adsorption processes [36]. Adsorption mechanisms of biofilm-covered plastic debris include physisorption, chemisorption and biosorption, involving the concernment of oxygen and nitrogen groups [37].

### 3.2. Adsorption Isotherms

After 45 days, the biofilm-covered plastic debris and virgin plastic debris were used to batch adsorption experiments. The fitting results of the Langmuir and Freundlich isotherms of Zn(II), Cd(II) and Pb(II) adsorption on PP, PET, biofilm-covered PP and biofilm-covered PET are seen in Figure 4 and Table 1. All the obtained experimental data were well fitted by the two isotherms, regardless the plastic types and metals. Meanwhile, the higher R^2^ suggested that the Langmuir modelis more fitted than the Freundlich model in this study, indicating that the metals adsorption by plastics may be dominant by monolayer adsorption [38].

For some plastic debris materials, the maximum adsorption capacity (Q_m_) for the three metals exhibited distinctly. For instance, the largest adsorption capacity was observed for Pb(II) on the plastic debris as 256.21 and 277.38 μg/g for biofilm-covered PP and biofilm-covered PET, respectively, in Xuanwu Lake; 250.90 μg/g, 334.38 μg/g for biofilm-covered PP and biofilm-covered PET, respectively, in Donghu Lake; 226.81 μg/g, 248.10 μg/g for biofilm-covered PP and biofilm-covered PET, respectively, in the Qinhuai River; whereas that for virgin PP and virgin PET was 176.35 μg/g and 216.51 μg/g, respectively. The lowest adsorption capacities were observed for Zn(II) among the three metals.

Trace metal adsorption capacity on plastic debris followed a similar trend as all biofilm covering is less on virgin plastic debris, indicating that the biofilms could obviously increase the trace metal adsorption capacity of plastic debris, consistent with previous studies. Additionally, there is no significant difference in the metal adsorption of biofilm-covered plastic debris in different waters.

The strong adsorption capacity of metal ion on biofilms was mainly due to the existence of the anionic functional groups, such as the carboxyl group and phosphate group of the cell walls [23]. Furthermore, EPS produced by the microorganism account of large organic components within biofilms, can increase the electrostatic interactions among metal ions and the biofilm-covered plastic debris. Besides, the metal adsorption on the biofilm-covered plastic debris also can be affected by the bioaccumulation, biotransformation and bio-adsorption [39].

### 3.3. Adsorption Kinetics

The curves of adsorption kinetics for Zn(II), Cd(II) and Pb(II) on PP, PET, biofilm-covered PP and biofilm-covered PET (using the pseudo-first-order and pseudo-second-order models) are shown in Figure 5 and Table 2. Accordingly, most adsorption capacity was attained within 12 h. After that, the metal adsorption reached a quasi-equilibrium.

Based on the R^2^ values, the pseudo-first-order model and the pseudo-second-order model fit the experimental data well (0.95 to 0.98). The equilibrium adsorption amount calculated by the pseudo-first-order model is lower than the obtained data and the pseudo-second-order model is more consistent with the experimental data than the pseudo-first-order model, except for virgin plastic debris. This indicates that the adsorption of metals may be mainly contributed to chemical adsorption. There is no significant difference in the metal adsorption capacity of biofilm-covered plastic debris in different waters. 

In the intraparticle diffusion model, the adsorption process is assumed to be preceded by a complex physicochemical mixed adsorption process, always used to study the rate-limiting step in the adsorption process [40]. As shown in Figure 6 and Table 3, the kinetic curves could be distributed into two or three linear segments, suggesting that various stages of adsorption occur during the adsorption reactions. The rate-limiting step is interparticle diffusion if the graph is linear and intersects the origin; otherwise, it is membrane diffusion [23]. 

As shown in Figure 6, the curves were nonlinear and did not transect the origin for biofilm-covered PP and PET, suggesting the adsorption may be limited by membrane diffusion. Furthermore, two stages can be seen from the metal adsorption on the virgin plastic debris, while three stages were observed for the adsorption on the biofilm-covered plastic debris. The initial linear segment represents membrane diffusion with fast adsorption rates; subsequent linear segments are related to intraparticle diffusion in the transitional part and the quasi-equilibrium phase. During the adsorption process, an increased C was observed, suggesting that the boundary-layer effect was enhanced, while, the decreased k_t_ during the adsorption process indicated that the diffusion rate slowed and the chemical adsorption reached quasi-equilibrium.

These results suggest that the adsorption kinetics of metals on plastic debris were significantly affected by the biofilms, by switching the intraparticle diffusion for virgin plastic debris to film diffusion for the biofilm-covered plastic debris. Previously, the physicochemical properties of plastic debris surfaces were altered by the biofilm formation, and the biofilm-covered plastic debris increased the metal adsorption through mechanisms such as electrostatic interaction and complexation with functional groups on the biofilm [11].

## 4. Conclusions

The adsorption of trace metals, i.e., Zn(II), Cd(II) and Pb(II), on plastic debris (PP, PET: virgin or biofilm-covered, in three aquatic systems) was studied. The SEM-EDS characterisation showed that a large amount of biomass was developed on the surface of biofilm-covered plastic debris, with the O element increased by 10 to 20% compared with the virgin plastics, which may strongly affect the ability to adsorb trace metals. The adsorption mechanisms of biofilm-covered plastic debris involving the participation of oxygen and nitrogen groups were demonstrated by FI-TR. The adsorption isotherms of trace metals on plastic debris were fitted by the Langmuir models. The adsorption capacities for Pb(II) were the highest of the different metals, at 256.21 μg/g and 277.38 μg/g for biofilm-covered PP and biofilm-covered PET, respectively, in Xuanwu Lake. For the three metals, the adsorption capacity on plastic debris followed the same trend of PET > PP and biofilm-covered plastic debris > virgin plastic debris. The adsorption kinetics of trace metals on plastics indicates that it may occur via chemical adsorption, except for virgin plastic debris. There is no significant difference in the metal adsorption capacity of biofilm-covered plastic debris in different waters. Intraparticle diffusion was involved in adsorption by virgin plastic debris, while adsorption by biofilm-covered plastic debris was governed by film diffusion. Thus, as an emerging anthropogenic adsorbent, plastic debris, especially biofilm-covered plastic debris, may exacerbate the transport of trace metals in freshwater environments. In addition, there is an increasing need to consider plastic debris and biofilm-covered plastic debris when assessing and simulating the transport and destiny of trace metals in freshwater. 

## Figures and Tables

**Figure 1 ijerph-19-13752-f001:**
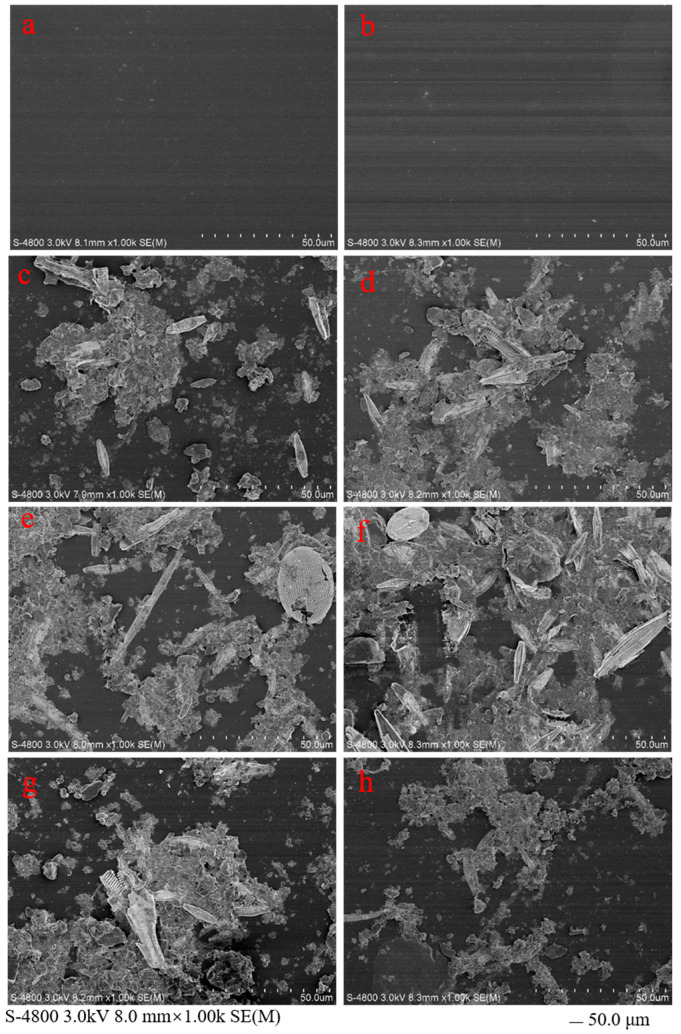
SEM images of plastic debris. (**a**) Virgin PP; (**b**) virgin PET; (**c**) biofilm-covered PP in Xuanwu Lake; (**d**) biofilm-covered PET in Xuanwu Lake; (**e**) biofilm-covered PP in Donghu Lake; (**f**) biofilm-covered PET in Donghu Lake; (**g**) biofilm-covered PP in Qinhuai River; and (**h**) biofilm-covered PET in Qinhuai River.

**Figure 2 ijerph-19-13752-f002:**
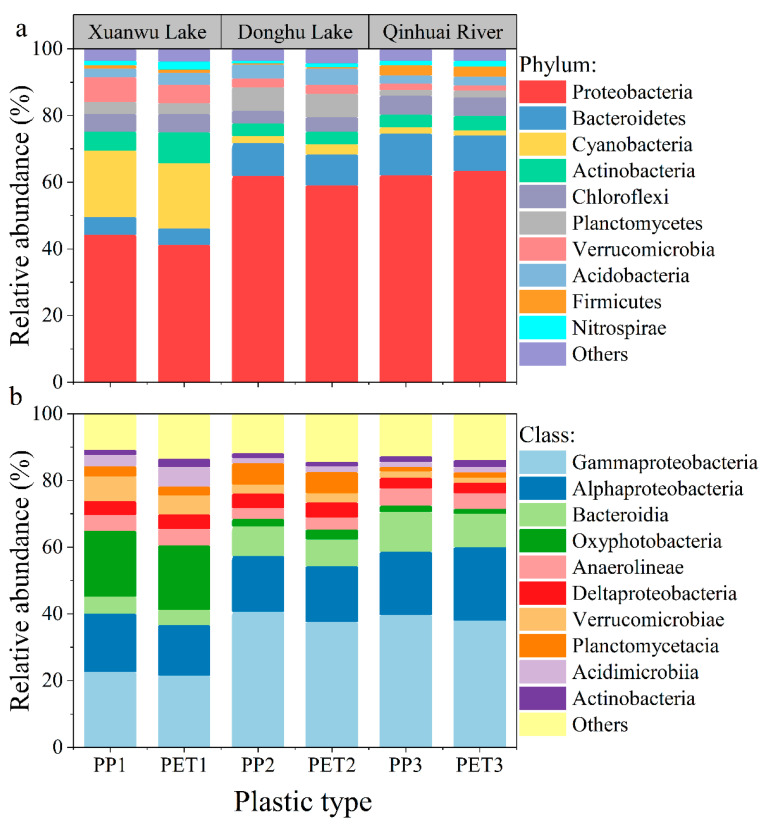
The abundance of bacteria at the phylum level (**a**) and class level (**b**) in biofilms on plastic debris. PP1 and PET1: biofilm-covered plastic in Xuanwu Lake; PP2 and PET2: biofilm-covered plastic in Donghu Lake; and PP3 and PET3: biofilm-covered plastic in the Qinhuai River.

**Figure 3 ijerph-19-13752-f003:**
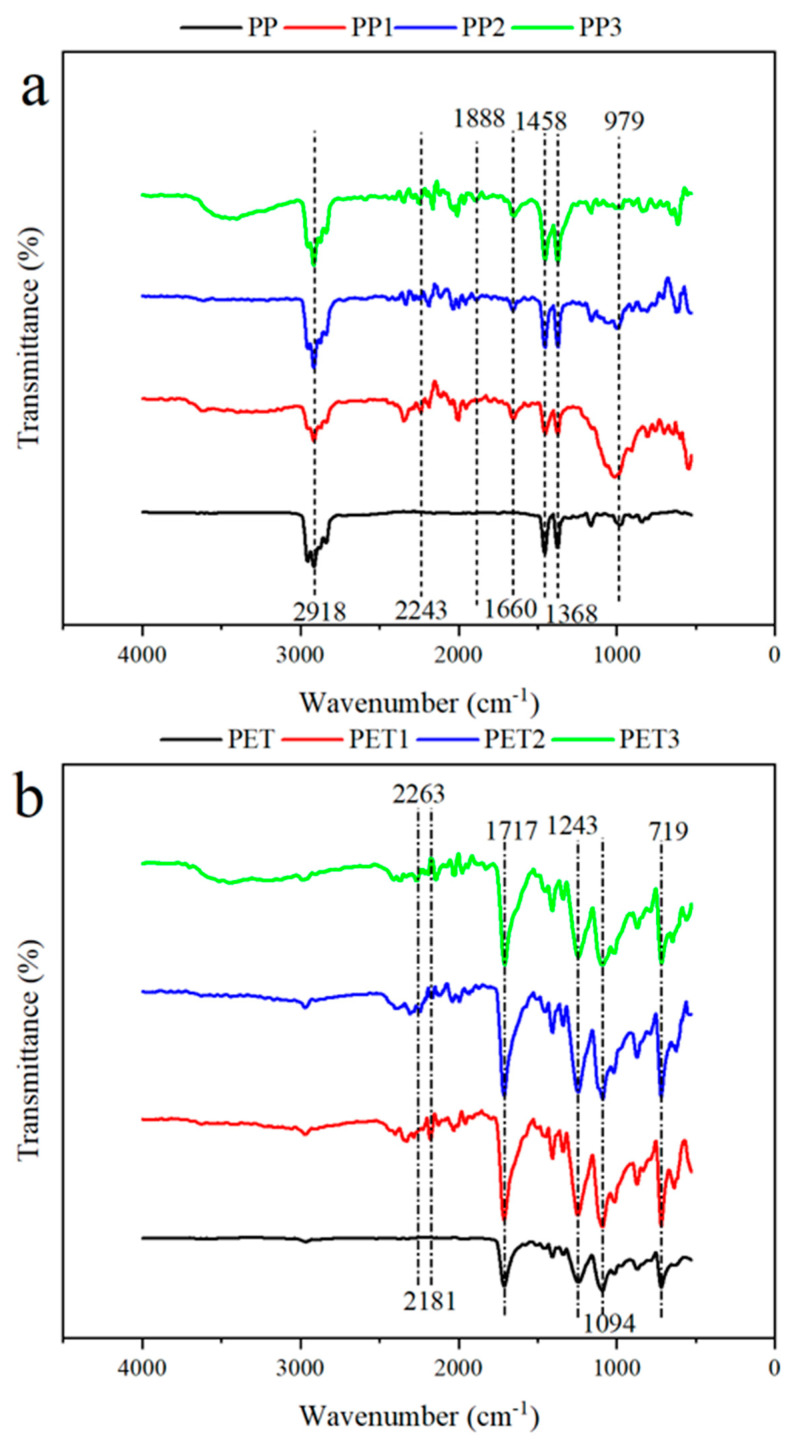
FT-IR of virgin and biofilm-covered plastic debris before adsorption, (**a**) for virgin PP and biofilm-covered PP; (**b**) for virgin PET and biofilm-covered PET. PP and PET: virgin plastic; PP1 and PET1: biofilm-covered plastic in Xuanwu Lake; PP2 and PET2: biofilm-covered plastic in Donghu Lake; PP3 and PET3: biofilm-covered plastic in the Qinhuai River.

**Figure 4 ijerph-19-13752-f004:**
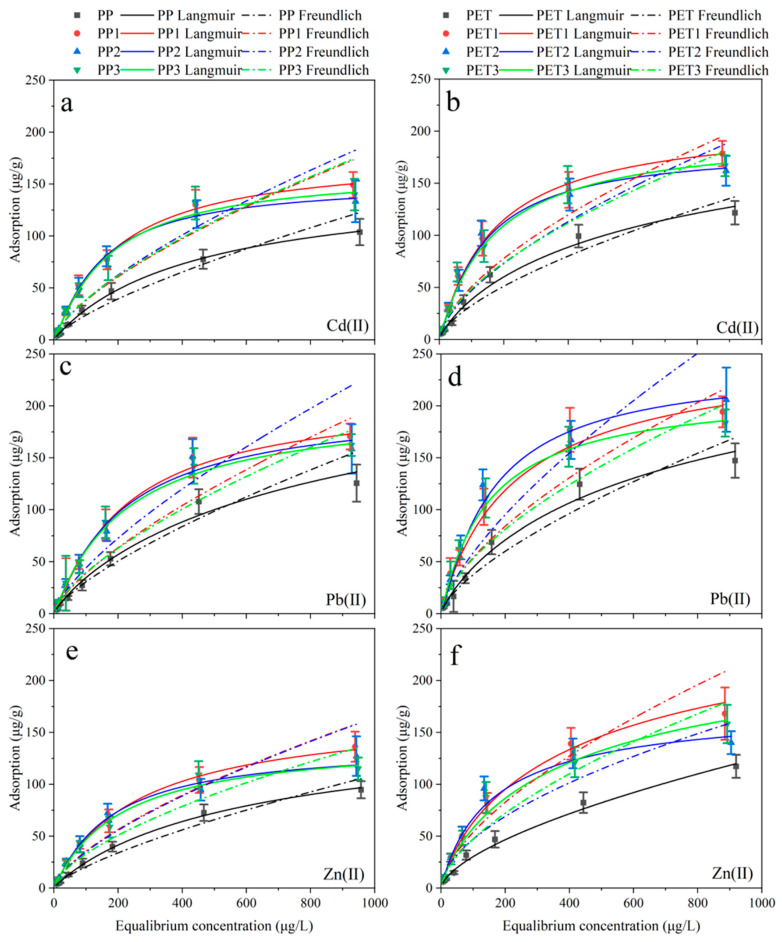
Adsorption isotherms of the Cd(II) (**a**,**b**), Pb(II) (**c**,**d**) and Zn(II) (**e**,**f**) for plastic debris. PP and PET: virgin plastic; PP1 and PET1: biofilm-covered plastic in Xuanwu Lake; PP2 and PET2: biofilm-covered plastic in Donghu Lake; PP3 and PET3: biofilm-covered plastic in the Qinhuai River.

**Figure 5 ijerph-19-13752-f005:**
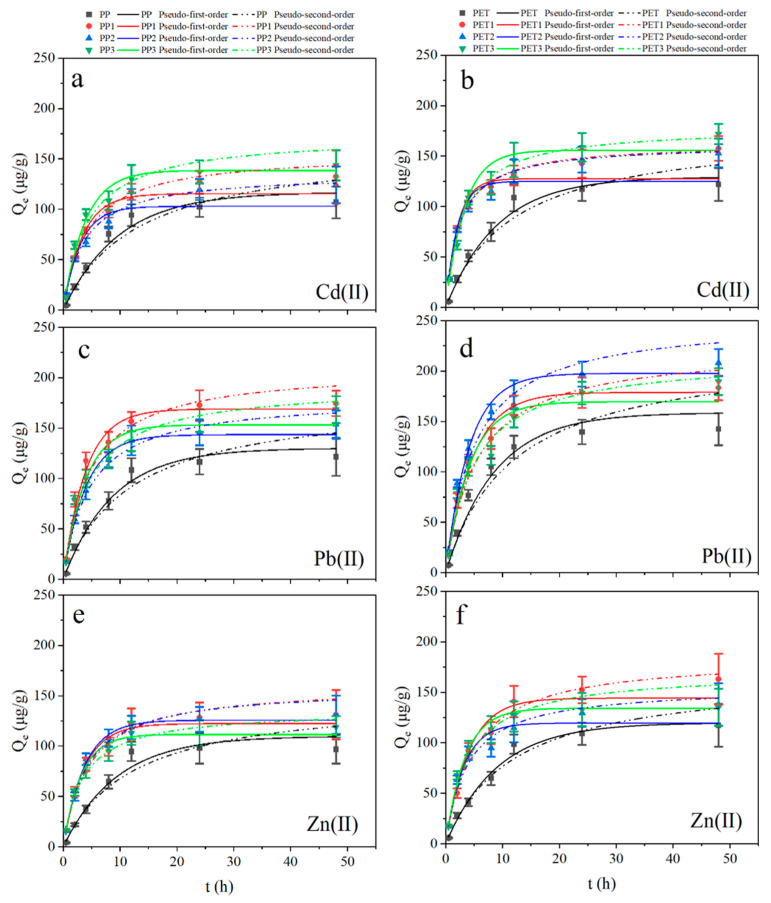
Adsorption kinetics of Cd(II) (**a**,**b**), Pb(II) (**c**,**d**) and Zn(II) (**e**,**f**) for plastic debris. PP and PET: virgin plastic; PP1 and PET1: biofilm-covered plastic in Xuanwu Lake; PP2 and PET2: biofilm-covered plastic in Donghu Lake; PP3 and PET3: biofilm-covered plastic in the Qinhuai River.

**Figure 6 ijerph-19-13752-f006:**
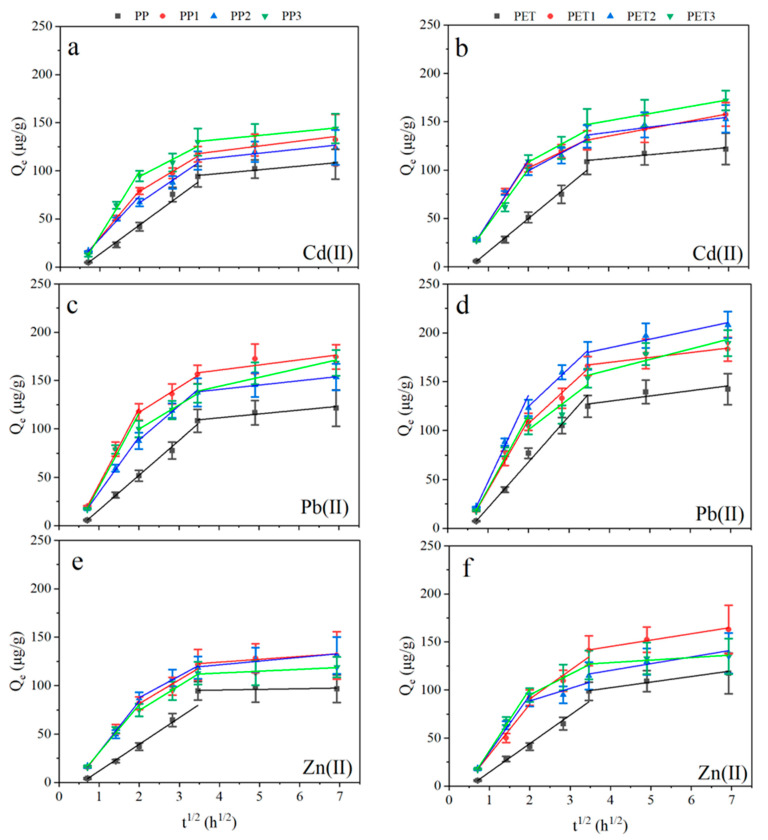
Intraparticle diffusion curves of Cd(II) (**a**,**b**), Pb(II) (**c**,**d**) and Zn(II) (**e**,**f**) for plastic debris. PP and PET: virgin plastic; PP1 and PET1: biofilm-covered plastic in Xuanwu Lake; PP2 and PET2: biofilm-covered plastic in Donghu Lake; PP3 and PET3: biofilm-covered plastic in the Qinhuai River.

**Table 1 ijerph-19-13752-t001:** Parameters of Langmuir adsorption isotherm and Freundlich adsorption isotherm models used in the studies.

Water	Plastic	Langumuir Isotherm									Freundlich Isotherm								
		Zn			Cd			Pb			Zn			Cd			Pb		
		Q_m_ (μg/g)	K_L_ (L/μg)	R^2^	Q_m_ (μg/g)	K_L_ (L/μg)	R^2^	Q_m_ (μg/g)	K_L_ (L/μg)	R^2^	K_F_ (L/μg)	1/n	R^2^	K_F_ (L/μg)	1/n	R^2^	K_F_ (L/μg)	1/n	R^2^
Virgin	PP	141.70	2.02 × 10^−3^	0.98	152.91	2.12 × 10^−3^	0.98	176.35	2.41 × 10^−3^	0.96	0.63	0.74	0.95	0.71	0.75	0.94	0.87	0.75	0.95
Virgin	PET	151.07	2.64 × 10^−3^	0.92	169.99	2.63 × 10^−3^	0.97	216.51	2.41 × 10^−3^	0.98	1.21	0.67	0.98	1.14	0.69	0.94	1.07	0.74	0.96
Xuanwu Lake	PP1	177.91	3.03 × 10^−3^	0.99	214.75	2.62 × 10^−3^	0.99	256.21	2.20 × 10^−3^	0.99	1.28	0.69	0.94	1.24	0.71	0.92	1.13	0.74	0.94
	PET1	225.26	3.00 × 10^−3^	0.98	247.04	2.75 × 10^−3^	0.98	277.38	2.20 × 10^−3^	0.99	1.67	0.70	0.95	1.67	0.69	0.93	1.62	0.70	0.94
Donghu Lake	PP2	162.69	3.29 × 10^−3^	0.99	202.13	2.80 × 10^−3^	0.98	250.90	2.38 × 10^−3^	0.99	1.21	0.70	0.92	1.19	0.73	0.89	1.17	0.76	0.91
	PET2	183.91	3.92 × 10^−3^	0.98	232.94	2.76 × 10^−3^	0.97	334.38	2.38 × 10^−3^	0.97	2.11	0.63	0.91	1.50	0.70	0.89	1.30	0.77	0.90
Qinhuai River	PP3	153.06	3.86 × 10^−3^	0.99	200.67	2.89 × 10^−3^	0.98	226.81	2.70 × 10^−3^	0.99	1.42	0.66	0.91	1.31	0.71	0.89	1.34	0.70	0.94
	PET3	203.84	3.20 × 10^−3^	0.97	228.35	2.97 × 10^−3^	0.98	248.10	2.70 × 10^−3^	0.99	1.63	0.68	0.94	1.76	0.67	0.93	1.80	0.68	0.92

PP and PET: virgin plastic; PP1 and PET1: biofilm-covered plastic in Xuanwu Lake; PP2 and PET2: biofilm-covered plastic in Donghu Lake; PP3 and PET3: biofilm-covered plastic in the Qinhuai River.

**Table 2 ijerph-19-13752-t002:** Parameters of pseudo-first-order model and pseudo-second-order model used in the studies.

Water	Plastic	Pseudo-First-Order									Pseudo-Second-Order								
		Zn			Cd			Pb			Zn			Cd			Pb		
		Q_e_ (μg/g)	K_1_ (1/h)	R^2^	Q_e_ (μg/g)	K_1_ (1/h)	R^2^	Q_e_ (μg/g)	K_1_ (1/h)	R^2^	Q_e_ (μg/g)	K_2_ (g/μg h)	R^2^	Q_e_ (μg/g)	K_2_ (g/μg h)	R^2^	Q_e_ (μg/g)	K_2_ (g/μg h)	R^2^
	PP	109.86	0.11	0.98	116.89	0.10	0.98	130.18	0.11	0.96	149.81	2.52 × 10^−4^	0.96	164.07	4.62 × 10^−4^	0.96	180.05	4.82 × 10^−4^	0.95
	PET	119.81	0.11	0.98	129.33	0.11	0.98	158.63	0.11	0.98	167.32	5.03 × 10^−4^	0.98	176.21	4.79 × 10^−4^	0.97	221.50	3.85 × 10^−4^	0.96
Xuanwu Lake	PP1	122.38	0.27	0.99	115.42	0.30	0.99	168.94	0.26	0.99	160.49	1.38 × 10^−3^	0.99	156.15	1.50 × 10^−3^	0.99	209.92	1.05 × 10^−3^	0.94
	PET1	114.30	0.25	0.98	127.62	0.48	0.98	175.63	0.21	0.99	185.05	1.12 × 10^−3^	0.99	163.06	2.52 × 10^−3^	0.99	223.86	8.13 × 10^−4^	0.94
Donghu Lake	PP2	122.80	0.27	0.99	102.90	0.33	0.98	143.56	0.26	0.99	158.04	1.54 × 10^−3^	0.99	135.63	2.02 × 10^−3^	0.99	180.73	1.21 × 10^−3^	0.91
	PET2	119.58	0.32	0.98	125.02	0.49	0.98	158.59	0.24	0.97	155.68	1.68 × 10^−3^	0.97	161.82	2.57 × 10^−3^	0.99	252.06	7.88 × 10^−4^	0.90
Qinhuai Lake	PP3	111.67	0.32	0.99	138.59	0.26	0.97	153.25	0.26	0.99	135.79	2.13 × 10^−3^	0.99	173.42	1.35 × 10^−3^	0.95	192.75	1.21 × 10^−3^	0.94
	PET3	134.23	0.30	0.99	155.75	0.31	0.98	183.96	0.24	0.99	170.44	1.45 × 10^−3^	0.96	178.76	1.84 × 10^−3^	0.98	213.79	9.50 × 10^−4^	0.92

PP and PET: virgin plastic; PP1 and PET1: biofilm-covered plastic in Xuanwu Lake; PP2 and PET2: biofilm-covered plastic in Donghu Lake; PP3 and PET3: biofilm-covered plastic in the Qinhuai River.

**Table 3 ijerph-19-13752-t003:** Parameters of intraparticle diffusion model used in the studies.

Trace Metals		Waters							
				Xuanwu Lake		Donghu Lake		Qinhuai River	
		PP	PET	PP1	PET1	PP2	PET2	PP3	PET3
Zn(Ⅱ)	kt1	27.51	29.68	52.93	52.48	52.57	59.72	49.33	64.43
	C	−15.64	−15.19	−21.83	−19.76	−21.20	−24.69	−18.32	−27.6
	R2	0.98	0.99	0.99	0.98	0.99	0.99	0.98	0.99
	kt2	0.68	5.83	24.67	29.87	22.39	13.10	25.55	21.49
	C	92.71	79.07	32.05	30.91	42.78	62.41	23.32	51.90
	R2	−0.05	0.93	0.95	0.88	0.98	0.57	0.99	0.99
	kt3			2.94	6.63	3.88	7.01	1.92	2.63
	C			112.5	118.78	105.97	92.39	105.31	117.95
	R2			0.74	0.95	0.85	0.84	0.77	0.72
Cd(Ⅱ)	kt1	30.22	34.12	49.1	63.21	44.55	64.08	67.25	56.88
	C	−16.73	−18.4	−18.65	−17.15	−15.25	−17.65	−34.74	−12.83
	R2	0.99	0.99	0.99	0.97	0.98	0.98	0.99	0.96
	kt2	3.80	3.80	24.88	20.25	28.06	22.09	21.39	22.47
	C	81.97	96.74	28.96	61.58	10.75	55.19	51.34	63.3
	R2	0.85	0.83	0.98	0.99	0.98	0.96	0.92	0.91
	kt3			5.67	7.71	6.31	5.27	4.03	7.24
	C			99.88	104.27	96.3	117.91	116.52	120
	R2			0.87	0.99	0.82	0.86	0.96	0.99
Pb(Ⅱ)	kt1	35.94	46.76	78.16	71.96	57.52	88.54	75.94	75.25
	C	−19.78	−25.62	−35.53	−32.99	−23.33	−41.00	−35.99	−34.86
	R2	0.99	0.99	0.99	0.99	0.99	0.98	0.99	0.97
	kt2	4.00	38.29	26.11	38.29	35.45	39.10	25.07	31.61
	C	95.43	108.55	64.71	30.47	17.51	46.42	49.52	37.57
	R2	0.87	0.59	0.98	0.95	0.99	0.96	0.95	0.75
	kt3			5.31	5.01	4.48	8.82	9.30	10.65
	C			139.59	149.77	122.85	149.39	106.79	119.58
	R2			0.69	0.85	0.98	0.84	0.86	0.82

## Data Availability

Data will be availability when requested.

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
