# Peer review of "Effects of Biofilms on Trace Metal Adsorption on Plastics in Freshwater Systems"

_ijerph, 2022, doi:10.3390/ijerph192113752_

Round 1

Reviewer 1 Report

Review for “Effects of biofilms on trace metal adsorption on plastics in freshwater systems” by Zhilin Liu et al.

The authors performed experiments to study the effects of biofilms on trace metal adsorption on plastics in freshwater systems.  However, there are some comments to be addressed before being considered for publication.

1.     The authors should be more careful with the grammar in the manuscript. For example, “Thus, there is a still need to investigate the effect …” should be “Thus, there is still a need to investigate the effect …”. Another example is on Page 6, “The biofilm-covered plastics were sliced into 5 mm pieces, placed 10 pieces into 100 mL centrifuge tube.”, which should be “Another example is on Page 6, “The biofilm-covered plastics were sliced into 5 mm x 5 mm pieces, and 10 pieces of them were placed into a 100 mL centrifuge tube.

2.     Please make the citation style consistent throughout the manuscript. I see numbers (e.g., 28, 29 on Page 8) and author names (e.g., (Huang et al., 2019; Zhou et al., 2020) on Page 8) existing at the same time. Please check the requirement of this Journal on the citation style.

3.     Please make the labels (a, b, c, …) of the figures consistent size. For example, the font size of the labels in Figure 1 is way too big, but it is way too small in Figure 4.

4.     The resolution of Figure 4-6 is too low to see the error bars. Please improve the quality of these figures.

5.     On Page 22, “ie Zn(II), Cd(II) and Pb(II)” should be “i.e., Zn(II), Cd(II) and Pb(II)”. Please check similar mistake in the other places of the manuscript.

6.     On Page 12, why does the “that” in “The infrared spectra of the PP plastic debris showed that the absorption peaks” have a different font style? Is this a mistake or do the authors want to emphasize something?

7.     On Page 12, the last paragraph (“Accordingly, both the C=O and …”) doesn’t have indentation as the other paragraphs do. Please add indentation. The same issue is identified for the last paragraph on Page 4 too. Please check other places for similar issues.

8.     The scale bars in the SEM images in Figure 1 are almost invisible. Please either improve the quality of the figures or add additional scale bars.

Reviewer 2 Report

This study focused on the metal adsorption capacity along with kinetic evaluation. This is an interesting topic. However, because the biofilms were formed by various microbes, metal adsorption was varied depending on the type of biofilm-forming microbes. So, the kinetic analysis should be meaningless. Then, for this, more experiments for characterizing biofilm-forming microbes and selecting good biofilm formers with high metal adsorption, which should be preceded.

Round 2

Reviewer 2 Report

This manuscript has been well revised according to the reviewer's comments.